# Image Guided Radiotherapy (IGRT) and Delta (Δ) Radiomics—An Urgent Alliance for the Front Line of the War against Head and Neck Cancers

**DOI:** 10.3390/diagnostics13122045

**Published:** 2023-06-13

**Authors:** Camil Ciprian Mireștean, Roxana Irina Iancu, Dragoș Petru Teodor Iancu

**Affiliations:** 1Department of Oncology and Radiotherapy, University of Medicine and Pharmacy Craiova, 200349 Craiova, Romania; 2Department of Surgery, Railways Clinical Hospital Iasi, 700506 Iași, Romania; 3Oral Pathology Department, “Gr. T. Popa” Faculty of Dental Medicine, University of Medicine and Pharmacy, 700115 Iași, Romania; 4Department of Clinical Laboratory, “St. Spiridon” Emergency Universitary Hospital, 700111 Iași, Romania; 5Oncology and Radiotherapy Department, Faculty of Medicine, “Gr. T. Popa” University of Medicine and Pharmacy, 700115 Iași, Romania; 6Department of Radiation Oncology, Regional Institute of Oncology, 700483 Iași, Romania

**Keywords:** radiomics, head and neck cancers, CBCT, radiotherapy, biomarker, predictive, CT, radio-chemotherapy

## Abstract

The identification of a biomarker that is response predictive could offer a solution for the stratification of the treatment of head and neck cancers (HNC) in the context of high recurrence rates, especially those associated with loco-regional failure. Delta (Δ) radiomics, a concept based on the variation of parameters extracted from medical imaging using artificial intelligence (AI) algorithms, demonstrates its potential as a predictive biomarker of treatment response in HNC. The concept of image-guided radiotherapy (IGRT), including computer tomography simulation (CT) and position control imaging with cone-beam-computed tomography (CBCT), now offers new perspectives for radiomics applied in radiotherapy. The use of Δ features of texture, shape, and size, both from the primary tumor and from the tumor-involved lymph nodes, demonstrates the best predictive accuracy. If, in the case of treatment response, promising Δ radiomics results could be obtained, even after 24 h from the start of treatment, for radiation-induced xerostomia, the evaluation of Δ radiomics in the middle of treatment could be recommended. The fused models (clinical and Δ radiomics) seem to offer benefits, both in comparison to the clinical model and to the radiomic model. The selection of patients who benefit from induction chemotherapy is underestimated in Δ radiomic studies and may be an unexplored territory with major potential. The advantage offered by “in house” simulation CT and CBCT favors the rapid implementation of Δ radiomics studies in radiotherapy departments. Positron emission tomography (PET)-CT Δ radiomics could guide the new concepts of dose escalation on radio-resistant sub-volumes based on radiobiological criteria, but also guide the “next level” of HNC adaptive radiotherapy (ART).

## 1. Introduction

Concurrent chemo-radiotherapy (CCRT) as a single treatment, or induction chemotherapy (IC) followed by CCRT, are part of the approach strategies for locally advanced head and neck squamous cell carcinoma (LA-HNSCC). The identification of a biomarker predictive of response could provide a solution for treatment stratification, in the context of high recurrence rates, especially for those associated with loco-regional recurrence. The proposed concept of radiomics involves extracting a large volume of data from medical imaging to improve diagnostic accuracy, but also to create prognostic and predictive models. We propose to identify arguments for the use of a new biomarker (variation Δ of radiomics features during treatment) predictive of the response of head and neck cancers to multimodal treatment by chemo-radiotherapy. The guiding of treatment based on a biomarker would have as a consequence the implementation of de-escalation strategies in order to limit toxicities or to escalate the treatment dose to increase the treatment response rate. Likewise, the prediction, by radiomics models, of the risk of toxicity, such as xerostomia, would have the consequence of replanning in order to limit doses to certain radiosensitive structures [1,2,3].

## 2. Aims and Scope

We proposed to synthesize, in a narrative review, the data that could be considered relevant for the implementation of the Δ radiomics concept, based on the variation of radiomics features between different moments of the acquisition of medical images for the purpose of treatment stratification. We performed a search in the MedLine database using the PubMed search interface with keywords delta, Δ, radiomics, head and neck cancers, HNC, HNSCC, biomarkers, and radiotherapy. The studies that were considered relevant were selected and analyzed and the data referring to radiomics (not featuring Δ variation), HNC diagnosis, or deep learning were excluded.

## 3. Radiomics—An Emerging Role in HNC

In the last decade, the development of image analysis processes and the rapid increase in the volume of high-quality medical data have facilitated the extraction of quantitative features from high-resolution medical imaging. The transformation of these images into data, invisible to the eye of the examiner, was called radiomics by Gillies and colleagues. The authors particularly note the potential of the method to facilitate clinical decisions and to improve the management of cancer patients. Even if radiomics is frequently used in clinical research and especially in oncology, the value of an analysis of feature variation (Δ-radiomics) during treatment should not be neglected, generally having pre-treatment and post-treatment images as references. If, in the early stages, the therapeutic success rate is relatively high, for LA-HNSCC cases treated by combining radiotherapy with Cisplatin or Anti-EGFR (epidermal growth factor receptor) Cetuximab, 40% of cases will not respond to treatment. After 2–3 years, 50–60% will present loco-regional recurrence, but even the rates of distant failure through metastasis of 20–30%, or 2–4% per year, of the second primary cannot be ignored [1,2,3,4].

Radiomics has two major advantages: it could be more specific, but it is also non-invasive. As a marker of tumor heterogeneity, radiomics has an advantage over biopsy due to the possibility of evaluating the tumor in toto. Whether in computer tomography (CT), magnetic resonance imaging (MRI), positron emission tomography (PET) or even ultrasonography (US), mammography, or digital radiographs (XRD), medical images are analyzed and extracted radiomic features could be used to build diagnostic, predictive or prognostic models [1,5].

Radiomics is therefore the ideal answer for a partnership between the concept of precision medicine and, in particular, of precision oncology and non-invasive biomarkers that could provide a digital tumor phenotype. Using mathematical algorithms, radiomics is based on certain steps, including the initial selection of images, the delimitation of the region of interest (ROI) for two-dimensional images or of the volumes of interest (VOI) for three-dimensional images selected manually, semi-automatically or automatically. Image processing and feature extraction, image analysis and the construction of a model are also essential steps of radiomics. Jha et al. consider that the digital tumor phenotype offered by radiomics will play a decisive role in precision oncology in the future. Radiomics could provide essential data not only about tumor heterogeneity, but also about the tumor microenvironment (TME) [6,7].

The same applications as in the case of radiomics, including differential diagnosis, prognosis biomarker, prediction of treatment response, but also those anticipating the risk of toxicities associated with the treatment, can be translated to the Δ radiomics concept. Based on recordings of tumor heterogeneity before and after treatment, Δ radiomics analysis could offer a new perspective for chemotherapy and radiotherapy response prediction in HNC. A systematic review, that aimed to identify Δ radiomics studies from Embase, PubMed, and ScienceDirect databases, identified forty-eight studies that corresponded to the selection criteria, of which six studies, representing 12.5%, are related to HNC. It should be noted that the large number of Δ radiomics studies is equal to the number of studies that analyze the radiomic feature dynamics in gastrointestinal cancers (other than rectal cancer) and in rectal cancer. Only in the case of lung cancer were a doubled number of Δ radiomics analysis studies identified compared to HNC [2,6,8,9].

## 4. The Multidisciplinary Approach in Locally Advanced HNC: Challenges in ERA Precision Oncology

More than 20 years ago, the benefit of the multimodal approach in locally advanced HNC was demonstrated. The prospective randomized multicenter trial led by Wendt and colleagues demonstrates a net benefit and higher loco-regional control rates at 3 years (36% vs. 17%) in favor of a concomitant chemotherapy regimen, based on Cisplatin, 5-Fluorouracil, and Leucovorin, versus radiotherapy alone. Furthermore, treatment-related toxicity rates were considered relatively high, especially in regard to grade 3 or 4 mucositis, which were associated with 38% of cases with concurrent chemotherapy treatment. In the study arm treated with radiotherapy alone, the rate of moderate and severe mucositis was reported as 17%. Treatment breaks or postponements were also significantly more associated with concurrent treatment due to hematological toxicity [10].

Thus, considering that treatment gaps could lead to a severe compromise of loco-regional control, especially if gaps occurred before day 19 after the start of therapy, identifying a balanced regimen that would obtain the best therapeutic ratio is still a challenge. A gap in treatment of 1 day is considered detrimental, but the effect is more intense for >3 days of interruption of radiotherapy. An equivalent dose in 2Gy fractions (EQD2), based on the linear quadratic model (LQ), is widely used to compensate for breaks during the radiotherapy treatment, but the differences in cell repopulation from one case to another make this strategy imprecise. Another method of compensating radiotherapy treatment gaps is the administration of two daily doses at an interval of at least 6 h. Both strategies could create an increase in the risk of toxicities by escalating doses received by organs at risk (OARs) [11,12,13,14].

One of the controversies is associated with the administration of the regimen of 40 mg/m^2^ doses of Cisplatin weekly versus Cisplatin at 100 mg/m^2^ given every 3 weeks, both delivered concurrently with radiotherapy. Without a doubt, a dose of 30 mg/m^2^ weekly has been proven to be inferior to a dose of 100 mg/m^2^, but weekly administration could bring a radio-sensitization benefit with improvements in the toxicity profile. By evaluating, a randomized controlled phase III multicenter study comparing the two regimens of chemotherapy plus definitive radiotherapy in a total dose of 70 Gy in 35 fractions for 7 weeks, delivered only by intensity modulated irradiation techniques in cases of HNC stage III and IV non-nasopharyngeal, the authors considered an irradiation dose of at least 60 Gy, and a cumulative Cisplatin dose of at least 200 mg/m^2^, as a complete treatment. A 5% (94% vs. 89%) higher treatment completion rate in favor of the arm that received weekly Cisplatin treatment, and a 3% higher treatment completion rate (80% vs. 77%) in favor of the same regimen, justifies the consideration of weekly Cisplatin as a standard regimen. Totals of 61.8% vs. 53.4% in loco-regional control at two years, again in favor of the weekly Cisplatin regimen, is also an argument supporting it. It is still necessary to mention that, considering that, especially in developed countries, there is a rapid increase in the incidence of HNC associated with human papillomavirus (HPV) infection, a different disease from HNC related to a long history of smoking, in India, where the study was carried out, the percentage of HNC cases associated with HPV infection is still relatively low (10–15%) [15,16,17,18].

Cetuximab remains a treatment option for head and neck squamous cell carcinoma (HNSCC) ineligible for platinum-based therapy. Even if the phase 3 randomized trial conducted by Bonner et al. demonstrates a net advantage in OS at 5 years versus radiotherapy alone, Cetuximab remains only a feasible alternative to Cisplatin, with a different toxicity profile from chemotherapy that induces a skin rash. A De-ESCALaTE and RTOG 1016 trial proposed treatment de-escalation for oropharyngeal cancer cases, but the use of Cetxumab instead of Cispaltin in concurrent radiotherapy was associated with a detriment in OS. Currently, a regimen based on Gemcitabine and Nedaplatin is evaluated as a radio-sensitizer in concurrent settings with radiotherapy, but the high rates of adverse effects make it necessary to adjust the doses [19,20,21,22,23].

Immune checkpoint inhibitors (ICI) entered the therapeutic spectrum of locally advanced disease in possible synergistic association with radiotherapy, the success of the PACIFIC trial having opened this horizon. Durvalumab has become the standard of care as a consolidation treatment for locally advanced non-small-cell lung cancers that have responded favorably to curative chemo-radiotherapy. Even if, in HNC, Pembrolizumab concurrent with radiotherapy was not associated with either a superior tumor control or a benefit in OS, the PembroRad regimen associates a more favorable toxicity profile for locally advanced HNSCC cases not eligible for Cisplatin [24,25].

Even if it is not considered standard treatment, induction chemotherapy that associates Cisplatin and 5-fluorouracil demonstrated a 93% overall response rate (ORR), similar results (86%) being reported by the Radiation Therapy Oncology Group (RTOG) trial, and both rates of response having been reported more than 30 years ago. Currently, a combination between platinum, taxanes, and fluorouracil (TPF) is considered to be the optimal treatment, with superior results in induction both compared to the platinum–fluorouracil regimen and compared to the platinum–taxane combination. Haddad et al. mention the benefit of the TPF regime in survival and in the preservation of the organ, being recommended both for cases with a poor prognosis and for reducing the risk of distant metastases or loco-regional failure in operable cases or those with a good prognosis [26,27,28].

## 5. Image-Guided Radiotherapy (IGRT)—The Cornerstone for Radiomics

Modern radiotherapy has evolved along with advances in the diagnosis and management of HNC. The proximity of radiosensitive structures and the need to limit treatment related toxicities that affect the quality of life (QoL), but also the treatment discontinuation risk with possible tumor control failure, are arguments that led to the implementation of more conformal irradiation techniques based on the ballistic localization of the tumor and the radiosensitive structures named organs at risk (OARs). If, for most of HNC anatomical sites, the implementation of intensity modulated radiotherapy (IMRT), based on the concept of image-guided radiotherapy (IGRT), brings a significant benefit in reducing xerostomia and dysphagia, in the case of nasopharyngeal cancer the implementation of the IMRT technique brings a net benefit in local tumor control over the older conformal technique. Not only the parotid irradiation but also the minor salivary glands and submandibular gland could be spared by IMRT irradiation, the limitation of radiation doses to these structures also being a factor associated with the reduction in moderate to severe xerostomia rates [29,30,31,32].

An improved dose conformity is also associated with steeper dose gradients, thus increasing the risk of the geographic missing of target volumes. Both the use of the CT simulator and the integrated on-board imaging (OBI) system, an imaging-based system that improves the patient’s radiation therapy setup accuracy, are included in the concept of IGRT. The displacement of internal organs between radiotherapy fractions, but also the position variation for target volumes and OARs during a treatment fraction, are the causes of possible uncertainties mentioned by Fernández-Rodríguez et al. [33].

On-board cone-beam-computed tomography (CBCT), for the alignment of head and neck cancer patients before radiotherapy, is a time-consuming method that also involves an additional radiation dose for the patient. A study that included 56 cases demonstrated that, without imaging guidance, 40.8% of the fractions would have been delivered at >5 mm distance from the target. However, by correcting the daily shift by using the first five CBCT displacement measurements, this percentage of fractions delivered outside the target could be reduced to 19%. Zumsteg et al. recommends, in this context, the use of daily CBCT to reduce the uncertainties related to the positioning of HNC patients for radiotherapy treatment. The concept of using CBCT, not only to ensure the accuracy of treatment delivery but also for early treatment response prediction, was implemented by Sumner et al. The authors propose the correlation between the 3 month follow-up imaging responses and overall survival (OS) of patients with HNSCC and oropharyngeal location, with volume changes evaluated by CBCT during radiotherapy. The proposed method was the measurement of the largest nodal conglomerate (LNC) at the time of CT simulation and also by using CBCT at the end of radiotherapy. A 30% reduction in the cranio-caudal dimensions of the tumor of the tumor-involved lymph node volume was correlated with improved OS, but the variation in the cranio-caudal dimensions was also correlated with the imaging response 3 months after treatment. The authors note the potential of CBCT imaging to be an early predictor of response for oropharyngeal cancer [34,35,36].

## 6. Δ Radiomics in HNC—From Image-Guided Radiotherapy (IGRT) and Adaptive Radiotherapy (ART) to Treatment Response/Toxicity Prediction

Tran et al. propose a non-invasive method, based on Δ radiomics, of quantitative ultrasound (QUS) extracted from metastatic lymph nodes in order to predict the therapeutic response in the first 24 h after the start of treatment with radical radiotherapy. Spectral and texture features extracted from QUS, at the beginning of radiotherapy and at 24 h, 1 week, and 4 weeks after the beginning of treatment, were evaluated in the study. By dividing the patients according to the response to treatment at 3 months into partial and complete responders, the authors build a predictive Δ radiomic model. A single-feature naïve-Bayes classification provides the best correlation with the therapeutic response. The predictive accuracies were 80, 86, and 85% at 24 h, 1 week, and 4 weeks, respectively. The authors note that, even if QUS Δ radiomics, evaluated at 1 week and 4 weeks, offers better results, the method also offers reasonable accuracy for early radiotherapy treatment response prediction [37].

Dividing a cohort of 93 HNC patients into a training (60 cases) and validation set (33 cases), Sellami and colleagues evaluated, in a Δ radiomic study, 88 features extracted from the gross tumor volume (GTV) delineated on CBCT images. After the selection by receiver operating characteristic (ROC) curves of those radiomic features significant for the weekly response to CCRT, those radiomic features that had a significant variation in at least 5% of the cases were evaluated. After these repeated exclusions of redundant radiomic features, only Coarseness was considered as a Δ radiomic feature predictor of response to chemo-radiotherapy. The association of clinical variables with the model based on Δ-radiomics variation could increase the accuracy of treatment response prediction. Thus, a combined model that included the hemoglobin value demonstrated predictive superiority over the Δ radiomic model [38].

Adaptive radiotherapy (ART) originated at the end of the 1990s, but Yan et al. highlight the concept for the first time in an article published in 2010. The National Radiotherapy Advisory Group in the UK has been recommending four-dimensional adaptive radiotherapy (4D-ART) as the future standard in radiotherapy since 2007, it being considered an “extremely encouraging and greatly promoting adaptive radiotherapy”. ART could contribute to improving radiotherapy patient outcomes and methods such as auto-segmentation, automated planning, and deformable registration algorithms are proposed as a response to anatomical and functional variations of the structures involved in treatment planning. HNC, lung, and cervix cancers are mentioned as potential beneficiaries of ART. Currently, MRI-guided online adaptive radiotherapy is considered as a method with possible major benefits for liver, pancreas, abdominal lymph nodes, and prostate cancers’ treatment planning. Weight loss and tumor response are considered essential factors associated with the need to correct the treatment plans. Avgousti et al. mentions, in an analysis that includes 85 articles, four major criteria advocating for ART: >10% weight loss; >1 cm deviation in external contour; >5% variations compared to the accepted doses; and variations of the planned doses in the periphery of the target volumes. The authors mention the value of ART for HNC radiotherapy and the ability of the method to decrease toxicity and improve local control. Hybrid adaptation, including image guidance combined with offline re-planning, is considered the optimal option to respond to organ motion and tumor shrinkage. Using fluoro-2-deoxyglucose (FGD) positron emission tomography (PET)-CT imaging, before and after CCRT treatment but also weekly after each chemotherapy session, for HNC cases, Yan and colleagues conceptually propose adaptive dose painting by number (DPbN) based on the voxel dose response matrix obtained by evaluating the voxel dose response matrix and the baseline standardized uptake value (SUV). The conclusions mention the existence of tumor voxels not correlated with baseline SUV as the main regions of radio-resistance. For HNC not related to human papillomavirus (HPV), radio-resistance tumor subvolumes are identified requiring >100 Gy in order to obtain adequate local control. DPbN is considered feasible as an adapted method to radiobiological heterogeneity in order to overcome radio-resistance [39,40,41,42,43,44].

The early prediction of the anatomical variations of the structures’ volume involved in HNC radiotherapy treatment could provide an indication of ART re-planning to avoid tumor coverage errors and radiation overdose of OARs. In order to identify those cases associated with major anatomical variation as a result of the early response to irradiation, a study based on the weekly assessment of CBCT used for IGRT was proposed. A total of 104 Δ-radiomics were extracted from the clinical target volume (CTV) and parotid gland volume. A model composed of 13 features extracted from CTV and 6 features extracted from parotid glands offered a 0.90 accuracy, 0.95 sensitivity, and 0.86 specificity performance for the previously mentioned end-point. The variation of three features, including the gray level matrix features family, were identified as significant in both analyzed structures [45].

An ability evaluation of radiomic features kinetics, extracted from daily mega-voltage CT (MVCT) scans for HNC radiotherapy IGRT used to predict moderate to severe xerostomia, was compared to the dose/volume parameters. The variation Δ of radiomic features extracted from MVCT for the contralateral parotid gland were evaluated as a predictor of moderate to severe xerostomia at 6, 12, and 24 months after treatment. The Δ radiomic model was compared with a pre-treatment dosimetric model and with a mixed dosimetric–radiomic model. For the models internal validation, cross-validation and bootstrapping were used. The Δ radiomic performances were quantified both on training sets and on test sets; area under the curve (AUC) values of 46%, 33%, and 26% for the moderate to severe xerostomia predictive rate at 6, 12, and 24 months, respectively, were obtained by testing the model. Radiomic Δ variations obtained from MVCT, comparing the baseline treatment’s half features, demonstrated superior predictive power compared to the pre-treatment dosimetric models. Pota et al. also mentions the use of the radiomic parameters’ dynamics, extracted from CT images until the halfway point of the treatment, as early predictors of radiotherapy-induced parotid shrinkage, but also for treatment-related toxicity. The fuzzy logic concept, a method of grouping data with the same characteristics into fuzzy sets and a model with the capacity of managing uncertain data and building rule-based classifiers, was proposed by the authors. Weekly CT changes during treatment quantified in Δ radiomic features were assessed on 68 treatment planning CTs and 340 weekly control CTs as a predictor of moderate to severe xerostomia 12 months after radiotherapy. Geometric, intensity, and texture features were evaluated for their predictive power in relation to the reference models based on mean radiation dose for contralateral parotid gland and nadir xerostomia score (Xerbaseline). The Δ variation of the contralateral parotid gland surface was the most predictive feature correlated with moderate and severe xerostomia at 12 months. The Δ changes of the surface of the parotid gland, 3 weeks after the treatment started, demonstrated superior predictive capacity compared to the pre-treatment models. QUS Δ radiomics obtained during CCRT is also proposed as a predictor of recurrence in HNSCC. A total of 31 spectral and texture radiomic features were extracted from a tumor-involved lymph node. The study group included 51 cases treated with a total radiotherapy dose of 70 Gy in 33 daily fractions of IMRT. The radiomic data were extracted before the treatment, and at one and four weeks after the beginning of the treatment. The study included patients with cancers of the oropharynx, larynx, hypopharynx, and of unknown primary; all cases were treated with platinum-based chemotherapy or Cetuximab. In a median follow-up of 38 months (range 7–64 months), relapse was identified in 17 cases. The model predicted the recurrence with an accuracy of 80% and 82% using Δ radiomics at one week and 4 weeks, respectively. The necessity of anticipating the result of chemo-radiotherapy treatment in HNC is also mentioned by Morgan et al., which also highlights the advantage of a response-adapted concept in order to escalate, de-escalate, or modify the treatment. In a retrospective analysis, the authors analyze, in a single institution study, the variation of radiomic features between baseline CT and CBCT, recorded at 1 day and 21 days during radiotherapy treatment. The analysis includes the primary tumor and metastases lymph nodes, with both local or lymph node failure being analyzed. The results were compared evaluating the clinical, radiomic model (including pre-treatment CT and radiomic Δ) and fused model (including both clinical and radiomic data). The fused model seems to offer superior predictive capacity for primary tumor (sensitivity of 78.3% and specificity of 90.9%), as well as for lymph nodes failure (sensitivity of 100.0% and specificity of 68.0%). Even if the study cannot demonstrate a net superiority of the radiomic and Δ radiomic algorithm compared to the clinical model, the fused model, including both clinical Δ radiomic data, is considered superior for the prediction of loco-regional failure including primary tumor, but also lymph nodes, relapse [46,47,48,49,50].

Evaluating the risk of acute xerostomia in a 35 patient lot with stage I-IVB nasopharyngeal cancer treated with IMRT radiotherapy, Liu et al. compared the model based on the variation Δ of CT radiomic features extracted from the parotid glands and a model based on saliva amount quantification. Radiomic features were extracted at fractions 0, 10, 20, and 30. Four patients were subsequently added to the lot for independent testing of the model. The delineation of the parotid gland was performed manually by the radiation oncologist, and the saliva was collected every 10 days during the treatment. Both the amount of saliva and the Radiation Therapy Oncology Group (RTOG) acute xerostomia score were used to quantify acute xerostomia. A total of 1703 radiomic features were extracted from each CT image of the parotid gland. RidgeCV and recursive feature elimination (RFE) methods were used for features selection and feature matrix reduction. Totals of 8 baseline radiomic features and 14 radiomic features extracted after 10 days of treatment were identified as significant for predicting acute xerostomia. The model that compared the variation of radiomic features between the baseline value and the value recorded at 10 treatment fractions offered the best predictive power, with a precision of 0.9220 and a sensitivity of 100% compared to the model based on saliva amount [51]. The main studies that evaluate Δ radiomics as a predictor of response to radiotherapy alone or in combination with systemic therapy, but also to anticipate early treatment related toxicity or the need for ART in HNC, have been summarized in Table 1, [35,37,38,45,46,47,48,49,50,51].

## 7. Conclusions

Δ radiomics demonstrates its potential as a predictive biomarker of treatment response in HNC and, conceptually, IGRT including CT simulation and CBCT positioning control imaging thus opens up new perspectives for the implementation of radiomics and Δ radiomics in radiation oncology. The use of Δ features of texture, shape, and size, both from the primary tumor and from the tumor-involved lymph nodes, demonstrates the best predictive accuracy. If, in the case of treatment response, promising Δ radiomics results could be obtained, even after 24 h from the start of treatment, for radiation-induced xerostomia the evaluation of Δ radiomics in the middle of treatment could be recommended. The fused models (clinical and Δ radiomics) seem to offer a benefit both in comparison to the clinical model and to the radiomic model. The selection of patients who benefit from induction chemotherapy is underestimated in Δ radiomic studies and may be an unexplored territory with major potential. The advantage offered by “in house” simulation CT and CBCT favors the rapid implementation of Δ radiomics studies in radiotherapy departments. PET-CT Δ radiomics could guide new concepts of dose escalation for radio-resistant sub-volumes based on radiobiological criteria, but could also guide the “next level” of HNC ART.

## Figures and Tables

**Table 1 diagnostics-13-02045-t001:** Δ Radiomics in HNC—from treatment response and toxicity to ART prediction.

Δ Radiomics in HNC			
Images Used for Radiomic Analysis	Number of Selected Features	Endpoint	References
Ultrasound	Single-feature naive-Bayes classification	Prediction of radiotherapy response at 3 months	Tran et al., 2020 [37]
CBCT	Two-dimensional (2D) maximum and craniocaudal (CC)	Prediction of radiotherapy response	Sumner at al., 2021 [35]
CBCT	1 feature (Coarseness); Clinical + radiomic model	Progression to radiotherapy for oropharingeal cancer	Sellami et al., 2022 [38]
CBCT	13 features extracted from CTV and 6 features extracted from parotid glands	Volume change in anatomical strictures; necessity for adaptive radiotherapy	Iliadou et al., 2022 [45]
MVCT	Combination of dose/volume and radiomics-based model	Prediction of moderate-to-severe xerostomia	Berger et al., 2022 [46]
CT	Texture features	Early prediction of radiotherapy-induced parotid shrinkage and toxicity during radiotherapy	Pota et al., 2017 [47]
Treatment planning CT and evaluation CT	Grey tone difference matrix (coarseness), kurtosis, and the median intensity	Prediction of late xerostomia	van Dijk et al., 2019 [48]
Ultrasound	31 spectral and related texture features	Prediction of higher risk of recurrence	Fatima et al., 2021 [49]
CT and CBCT	102 features including 18 first order, 14 shape, 24 gray level co-occurrence matrix (GLCM), 16 gray level run length matrix (GLRLM), 16 gray level size zone matrix (GLSZM), and 14 gray level dependence matrix (GLDM) features.	Prediction of local failure	Morgan et al., 2021 [50]
CT	Saliva amount and 1703 radiomics features based model	Prediction of acute xerostomia during radiation therapy for nasopharyngeal cancer	Liu et al., 2019 [51]

## Data Availability

Not applicable.

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
