# Peer review of "Image Guided Radiotherapy (IGRT) and Delta (Δ) Radiomics—An Urgent Alliance for the Front Line of the War against Head and Neck Cancers"

_diagnostics, 2023, doi:10.3390/diagnostics13122045_

Round 1

Reviewer 1 Report

The review proposed by Mirestean et al. is of interest to the scientific and clinical assembly. Radiomic associated with AI is the future of the plan of treatment, in order to predict the response to radiotherapy. HNSCC cancers, with poor prognosis, are especially concerned by this kind of development.

I would have one minor concern about this review. Authors should explain more precisely how they perform their systematic review (PRISMA guidelines ? which keywords ? wich process is used for the selection and exclusion of studies ?).

Author Response

Dear reviewer,
Thank you for the effort and time given to evaluate the mansucis and also for the positive appreciation. We introduced a paragraph specifying the type of review (narrative review) and mentioning the details of the inclusion and exclusion of the analyzed studies.
Kind Regards,
Camil Mirestean

Reviewer 2 Report

Well written review, confirming increasing role of Radiomics in evaluating different prognostic factors in Oncological patients treated with Radiotherapy. The use of CBCT in patient's positioning is a simple and available tool not only in HNSCC but also in lung and rectal cancer. Results of those studies seem promising and worth of further enquire.It is worth of publication.

Author Response

Dear reviewer,
Thank you for the effort and time given to evaluate the mansucis and also for the positive appreciation. At the suggestion of another reviewer, we introduced a paragraph specifying the type of review (narrative review) and mentioning the details of the inclusion and exclusion of the analyzed studies.
Kind Regards,
Camille Mirestean